# Forecasting Maximum Mechanism Temperature in Advanced Technology Microwave Sounder (ATMS) Data Using a Long Short-Term Memory (LSTM) Neural Network

**Warren Dean Porter [1,\*]** , **Banghua Yan [2]** and **Ninghai Sun [3]**

1  Science Systems and Applications, Inc., Lanham, MD 20706, USA
2  Satellite Meteorology and Climatology Division, STAR, NOAA, College Park, MD 20740, USA
3  Global Science Technologies, Inc., Greenbelt, MD 20770, USA
\*  Correspondence: warren.porter@noaa.gov; Tel.: +1-301-683-3509

**Abstract:** Among the monitored telemetry raw data record (RDR) parameters with the STAR Integrated/Validation System (ICVS), the Advanced Technology Microwave Sounder (ATMS) scan motor mechanism temperature is especially important because the instrument might be unavoidably damaged if the mechanism temperature exceeds 50 °C. In the current operational flight processing software, the instrument automatically enters safe mode and stops collecting scientific data whenever the mechanism temperature exceeds 40 °C. This approach inevitably leads to the instrument entering safe mode unnecessarily at a premature time, causing the loss of scientific data before the mechanism temperature reaches 50 °C. This study seeks to leverage the influence the main motor current, compensation motor current, and main motor loop integral error have on mechanism temperature to forecast the maximum mechanism temperature over the upcoming 6 min. A long short-term memory (LSTM) neural network predicts maximum mechanism temperature using ATMS RDR telemetry data as the input. The performance of the LSTM is compared with observed maximum mechanism temperatures by applying the LSTM coefficients to several cases. In all cases studied, the mean average error (MAE) of the forecast remained under 1.1 °C, and the correlation between forecasts and measurements remained above 0.96. These forecasts of maximum mechanism temperature are expected to be able to provide information on when the ATMS instrument should enter safe mode without needlessly losing valuable data for the ATMS flight operational team.

**Keywords:** Joint Polar Satellite System (JPSS); Advanced Technology Microwave Sounder (ATMS); long short-term memory (LSTM) networks; machine learning (ML); anomaly forecasting; scan drive mechanism temperature

## 1. Introduction

The STAR Integrated Calibration/Validation (or Cal/Val) System (ICVS) Long Term Monitoring (LTM) (hereinafter named ICVS for clarity) group has functioned for over a decade. Its mission is to provide the general public with accurate and informative monitoring that characterizes the quality of observations primarily from a series of NOAA satellites and/or instruments, including NOAA legacy Polar-orbiting Operational Environmental Satellites (POES) and Joint Polar Satellite System (JPSS) instruments. Notably, the quality of instrument parameters monitored by ICVS has increased significantly [1–7]. In particular, the instruments under the purview of ICVS include but are not limited to Advanced Technology Microwave Sounder (ATMS) [8], Cross-track Infrared Sounder (CrIS) [9], Visible Infrared Imaging Radiometer Suite (VIIRS) [10], Ozone Mapping Profiler Suite (OMPS) [11,12], and Advanced Microwave Sounding Unit-A (AMSU-A) [13]. Apart from AMSU-A, the abovementioned instruments fly aboard Suomi National Polar-orbiting Partnership (S-NPP) and JPSS-1, which is also known as NOAA-20. In addition, these four instruments will also fly aboard JPSS-2 after its launch, which is currently planned for late

2022. The monitoring parameters cover more than 4000 instrument-based raw data record (RDR) and sensor data record (SDR) observations, thus offering significant information regarding the health of JPSS instruments and quality scientific data for satellite flight team engineers, calibration scientists, and data users within NOAA and beyond. A detailed introduction to the fundamentals of the ICVS LTM and related advanced inter-sensor functions can be found in recent studies [14,15].

Among the instruments monitored by ICVS, the ATMS is a cross-track scanning microwave instrument that measures profiles of atmospheric temperature and moisture. One key feature of microwave sounders such as the ATMS is their ability to make observations under all weather conditions since the microwaves emitted at various levels of the atmosphere are not inhibited by clouds as they travel to the instrument. Combining the capabilities of its predecessors, the AMSU-A and the Microwave Humidity Sounder (MHS), the ATMS provides observations of antenna (brightness) temperatures at 22 channels ranging from 23 to 183 GHz. Previous studies document the immense value the ATMS SDRs provide for short-to-medium range (1–10 days) weather forecasts generated by numerical weather prediction (NWP) systems [16–19]. Various satellite-based algorithms use these data to retrieve environmental data records (EDRs) describing processes of the Earth's atmosphere, ocean, and surface [20–28]. ICVS also uses machine learning to better characterize JPSS data [29,30].

For all aforementioned invaluable applications, a continuous supply of scientifically valid SDRs from ATMSs is of the utmost importance. However, the instrument does not collect such data when placed in its "safe mode". Safe mode events protect valuable instrument components against damage caused by overheating by disabling certain functions of the instrument, thereby allowing it to cool down (Figure 1). In the case of S-NPP ATMS, one source of potentially damaging excessive temperatures is the scan drive mechanism, which is a single-axis torque-compensating continuous-rotation gimbal assembly [31]. Permanent damage to the ATMS instrument can occur whenever the temperature of the gimbal assembly in the scan drive mechanism rises above 50 °C. Accordingly, it is important to predict these overheating events with enough forewarning to allow a switch to safe mode operations before the mechanism heats up enough for permanent damage to occur. Likewise, it is important that such predictions do not trigger safe mode needlessly, causing unnecessary interruptions to the collection of scientifically valid ATMS SDRs. This study seeks to improve upon existing methods and discover whether it is possible to predict spikes in mechanism temperature, i.e., a local maximum mechanism temperature, using observations of various parameters found in ATMS telemetry RDRs in a long short-term memory (LSTM) artificial neural network (ANN). In short, the current status quo consists of the following shortcomings in capabilities:

1. Limited use of deep learning methods [32] (including LSTM) to retroactively classify telemetry data anomalies [33–36];
2. No attempts to predict increases in mechanism temperature using any method;
3. Transition to the ATMS' safe mode operations is controlled only by whether current observations exceed a threshold value and not guided by any attempted forecasts.

Based on our best knowledge, this study is the first published attempt to go beyond these gaps. This is the first application of an ANN to predict a local maximum mechanism temperature for the JPSS ATMS instrument. Making such predictions in near real time (NRT) facilitates improvements in ATMS operation. Accordingly, this study seeks to describe a novel method for reducing data loss in an operational environment when the scan drive mechanism temperature exceeds the threshold at which safe mode has historically been activated. Likewise, this is a new application of the LSTM method that demonstrates the ever-increasing versatility of neural networks in a variety of tasks.

This study is organized as follows: The next section introduces LSTMs and presents the design of LSTM in a model that predicts maximum mechanism temperature. The prediction of ATMS maximum mechanism temperature using LSTM is given in the following section. A summary and conclusions are provided in the final section.

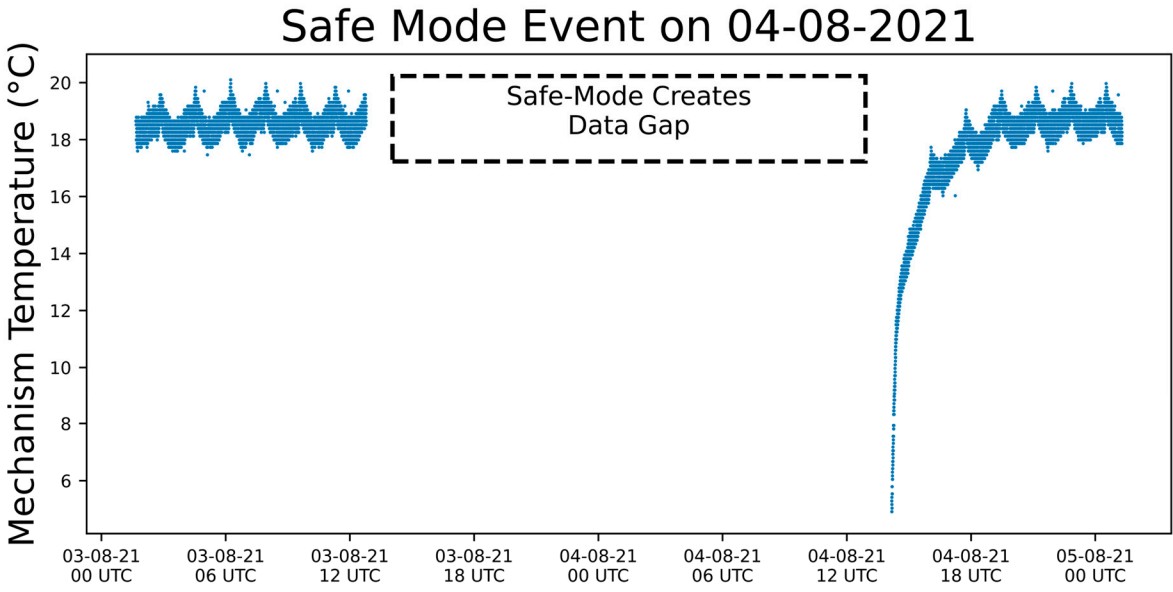

**Figure 1.** Time series of mechanism temperature during a safe mode event in August 2021. No valid data was collected when the instrument was in safe mode, resulting in a notable data gap. In addition, the mechanism temperature dropped tremendously during safe mode.

## 2. Long Short-Term Memory (LSTM) Method and Its Application for Predicting Maximum Mechanism Temperature

### 2.1. An Introduction to Long Short-Term Memory (LSTM) Networks

Like other recurrent neural networks (RNNs), LSTMs provide an advantage over other deep learning methods because they contain a "memory" capable of learning connections between observations in the past and those in the immediate future. This has made RNNs an attractive means for unsupervised anomaly detection [32], particularly when it comes to satellite spacecraft [33–36]. Unlike other RNNS, LSTM's architecture avoids vanishing and exploding gradient problems [37]. This makes LSTM an excellent tool for predicting local maxima in a time series [38], such as the mechanism temperature (*T*) [39]. A brief summary of LSTM networks is given below, but a more detailed description can be found in its original publication [40].

The LSTM layer is composed of a series of sequentially connected cells (Figure 2) rather than an array of nodes connected with nodes in other layers but not directly connected to each other. A detailed description of one cell is shown in Figure 3. As shown in the figure, a cell at timestamp *t* receives three inputs, two from the cell at the preceding timestep, cell state ($C_{t-1}$) and hidden state ($h_{t-1}$), and one from the input layer of the neural network at the current timestep ($x_t$). The cell state serves as the "memory" of the LSTM, while the hidden state is the output that will be used to make predictions and provide additional information to the succeeding cell. Both the cell state and hidden state are vectors with the same dimensionality.

Like other neural networks, an LSTM unit computes its outputs using a series of trainable weights and biases. Each cell in the LSTM contains four gates, each with an identical number of these weights and biases. For notation purposes, weights acting upon the input vector are denoted with a *W,* and weights acting upon the hidden vector are denoted by *U*. All four of these gates take the exact same input, a concatenation of the input vector from the current timestep ($x_t$) and the hidden vector from the previous timestep ($h_{t-1}$). Each gate contains an internal deeply connected layer that applies a linear transformation to the concatenated vector.

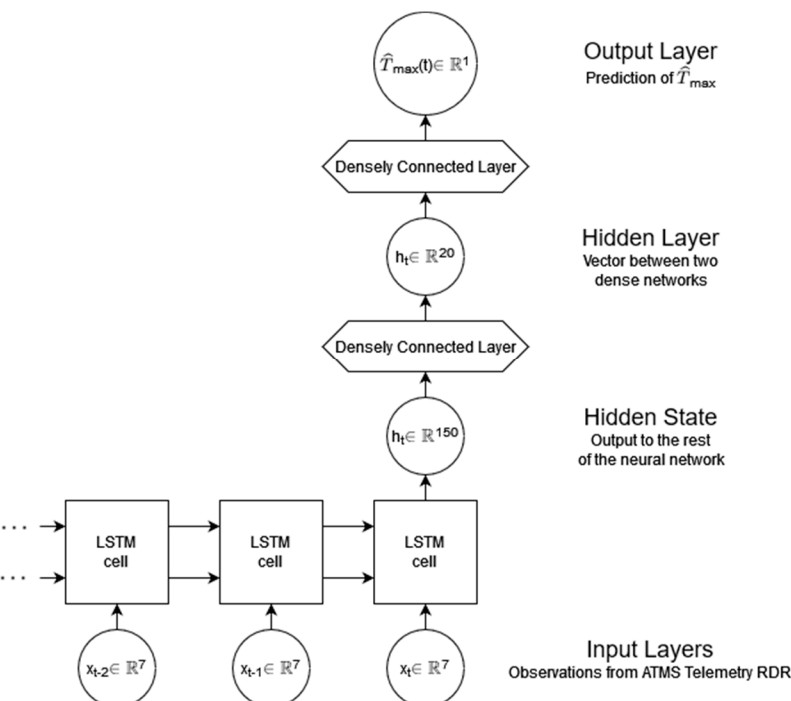

**Figure 2.** Simplified schematic showing how a series of sequentially arranged LSTM cells are connected. A prediction ($\hat{T}_{max}$) is created by passing the output of a single LSTM cell through two deeply connected layers. The LSTM output is a function of the most recent input vector and the two output vectors produced by the previous LSTM cell in the temporal sequence.

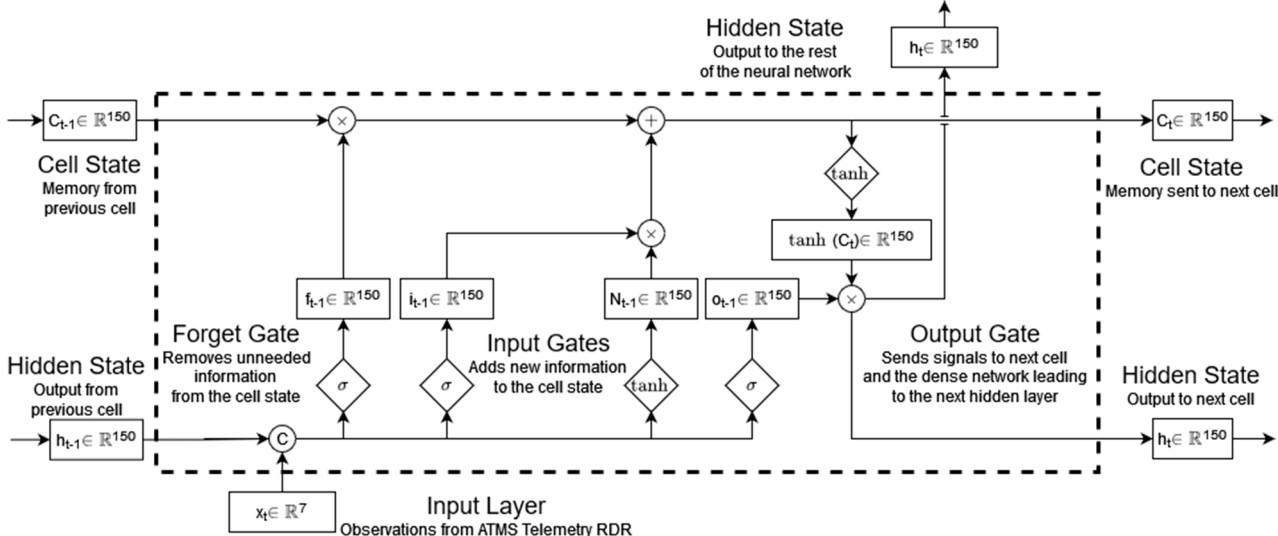

**Figure 3.** Schematic of an LSTM cell. Circles with a + or × represent piecewise matrix operations. Circles with a C represent vector concatenation. Seven inputs taken from ATMS Telemetry RDRs form the vector $x_t$ at each time step, which enters the LSTM at the bottom of the schematic.

One gate that processes the concatenated vector is the forget gate, which computes the forget matrix:

$$f_t = \sigma\left(W_f\,x_t + U_f\,h_{t-1} + b_f\right), \tag{1}$$

Linear multiplication of the weights $W_f$ and $U_f$ with $x_t$ and $h_{t-1}$, respectively, leads to a matrix that can be an input to the sigmoid activation function ($\sigma_f$). The sigmoid activation function is nonlinear and forces the forget matrix to only contain values in the open interval (0, 1). When the forget matrix is multiplied piecewise with the cell state from the previous

step ($C_{t-1}$), any values in the previous cell state that correspond to values near zero in the forget matrix are reduced to near zero. This near-removal of a value is how the forget matrix causes the cell state to "forget" information that may not be relevant any longer.

The next two gates are the input gates, which form a matrix from the piecewise multiplication of two separate matrices, each created from a separate gate:

$$i_t = \sigma \left( W_i \, x_t + U_i \, h_{t-1} + b_i \right), \tag{2}$$

$$\check{C}_t = \tanh \left( W_c \, x_t + U_c \, h_{t-1} + b_c \right), \tag{3}$$

The first of these matrices is the input matrix, $i_t$, created by a similar process to the formation of the forget matrix, albeit with separately trained weights and biases applied to the concatenation of $x_t$ and $h_{t-1}$. The second of these matrices, $\check{C}_t$, is also created by funneling the concatenation of $x_t$ and $h_{t-1}$ through yet another sequence of separately trained biases and weights, but the hyperbolic tangent (tanh) serves as the activation function instead of the sigmoid. Because tanh outputs to the interval $(-1, 1)$ rather than $(0, 1)$, it allows negative values to be included in the matrix that updates the cell state. Unlike the forget matrix, which is ultimately multiplied piecewise with the cell state, the input matrix undergoes piecewise addition with the cell state, which allows for new information to be added to the cell state and affect the output of subsequent cells.

$$C_t = f_t * C_{t-1} + i_t * \check{C}_t, \tag{4}$$

Thus, the cell state is a function of the previous cell state, with the forget matrix removing unneeded information from the previous cell state and the input matrix adding new information to the cell state. In addition to becoming an input for the subsequent LSTM cell, $C_t$ is an input to operations affecting the cell's output, $h_t$.

The final gate is the output gate, which creates the hidden vector that feeds both the LSTM cell at the next timestep as well as the deeply connected layers that ultimately forecast the relevant predictand. Here, $h_t$ is created from the piecewise multiplication of two separate matrices. One is the previously computed new cell state activated by the hyperbolic tangent function, $\tanh(C_t)$, and the other, $o_t$, is the result of yet another gate operating upon the concatenated vector, $x_t + h_{t-1}$. Once again, a deeply connected network architecture with separately trained weights and biases supplies the necessary linear transformation.

$$o_t = \sigma \left( W_o \, x_t + U_o \, h_{t-1} + b_o \right), \tag{5}$$

$$h_t = o_t * \tanh \left( C_t \right), \tag{6}$$

Therefore, $h_t$ is a function of both the current input vector, $x_t$, previous cell state, $c_{t-1}$, and the results from the previous cell, $h_{t-1}$. A key feature of the LSTM architecture is the separation of the weights given to the four gates, keeping the processes of memory deletion, memory addition, and output computation segregated.

### 2.2. LSTM Design for Forecasting Local Mechanism Temperature Maxima

To apply the abovementioned LSTM to ATMS data for forecasting local mechanism temperature maxima, each cell of the LSTM corresponds to a single timestep with 8 s, while the size of the vectors passed from one cell to the next is set to 150 nodes. Here, each node contains one element of the vector. The vector size of 150 elements is determined based on the sensitivity analysis for 5 ATMS anomaly events about the mean retrieval error, as shown in Figure 4. The MAE decreases rapidly once the size of the cell memory reaches at least 100. Because the MAE is insensitive to further increases in memory, this study utilized 150 nodes in the cell memory of its LSTM model.

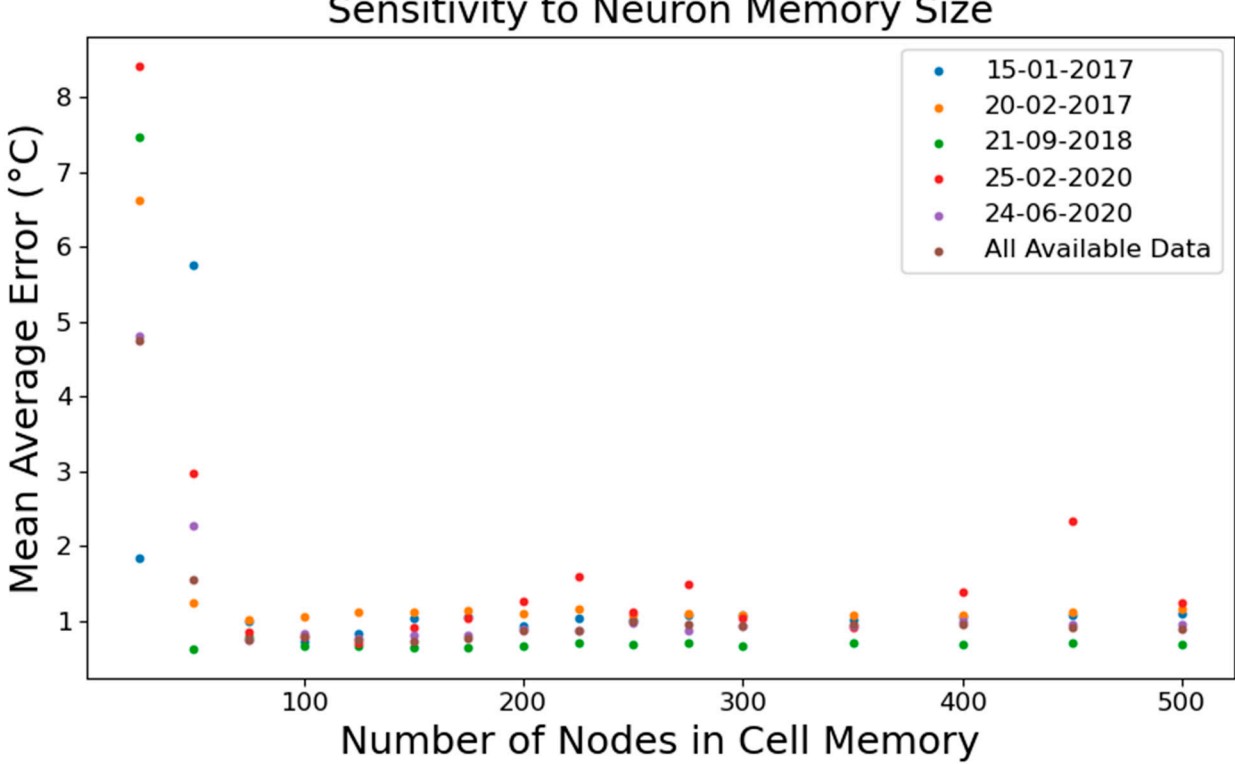

**Figure 4.** The MAE for each case is insensitive to the size of the internal cell state and hidden layer vectors. In nearly all cases, MAE is around 1 °C or lower.

As shown in Figures 2 and 3 above, in the LSTM, the 150-element vector gets fed into a dense layer that transforms it into a 20-element vector. Another dense layer transforms the 20-element vector into a 1-element vector, which is $\hat{T}_{max}$, the forecast of maximum mechanism temperature, $T$. The temperature $\hat{T}_{max}$ is a predictand that can be used to control whether ATMS enters safe mode or not. These final two dense layers function identically to the deeply connected layers of an ANN. This part of the diagram is also shown in Figures 2 and 3 above.

The LSTM computes its output, a 150-element vector in this study, by passing a concatenation of current observations and previous outputs through four gates. In this study, the input vector at the current timestep contains 7 elements, and the hidden vector from the previous timestep contains 150 elements; therefore, the concatenated vector that enters the four gates in each cell contains 157 elements. Each of the four gates consists of a dense layer with $157 \times 150 + 150 = 23{,}700$ trainable parameters. These parameters include $157 \times 150$ weights connecting the 157 nodes in the concatenated vector with 150 nodes in the hidden vector and an additional 150 biases. With four such gates in each LSTM cell, the total number of trainable parameters is $4 \times 23{,}700 = 94{,}800$ weights and biases for each cell in the whole LSTM network. In this study, each cell's weights and biases are identical, so the number of cells does not affect the total number of trainable parameters.

In addition, generating forecasts of $\hat{T}_{max}$ requires chaining multiple LSTM cells to work together in temporal space (Figure 2 above). The cell corresponding to time $t-1$ passes its cell state ($c_{t-1} \in \mathbb{R}^{150}$) and its hidden state ($h_{t-1} \in \mathbb{R}^{150}$) to the next cell, which corresponds to time $t$. The cell at time $t$ combines the cell state and hidden state it receives from the previous cell with the most recent telemetry observations ($x_t \in \mathbb{R}^7$) according to the weights and biases in its four internal gates (Figure 3), which were described in detail above. This produces a new hidden state ($h_t \in \mathbb{R}^{150}$) that is outputted to the next layer, which is a dense layer deeply connected to the previous LSTM layer and a successive dense layer. The first dense layer transforms $h_t \in \mathbb{R}^{150}$ into $v_t \in \mathbb{R}^{20}$, reducing the dimensionality

considerably. The second dense layer transforms $v_t \in \mathbb{R}^{20}$ into $\hat{T}_{max}(t) \in \mathbb{R}^1$, which is the prediction of maximum mechanism temperature.

Implementation of the LSTM architecture relies upon several open-source software libraries. Python [41] serves as the primary programming language because of the flexibility offered by its packages, including NumPy [42], Pandas [43], and Matplotlib [44], which process, manage, and visualize data, respectively. The Keras [45] interface to the Tensor-Flow [46] library developed by Google facilitates the creation of many ANNs, including RNNs such as the LSTM.

## 3. Prediction of ATMS Maximum Mechanism Temperature Using LSTM

### 3.1. Acquisition of datasets for Testing and Training and Determination of the Optimal Duration for Predicting Maximum Mechanism Temperature

The plausibility of predicting spikes in mechanism temperature first requires the definition of a useful predictand. Qualitatively as shown in a time series of every parameter for an anomalous ATMS event in the S-NPP ATMS telemetry RDR (Figure 5), the current from the main motor, $I_{mm}$, the current from the compensator motor, $I_{cm}$, and the error in the main motor loop integral, $E_{li}$, are correlated with future spikes in the overall temperature of the mechanism, $T$, a few minutes later. Apparently, the main motor current and loop integral error increase a few minutes before mechanism temperature ($T$). This pattern appears in all the anomalies examined in this study. Because overheating events are the only relevant issue at hand, it is more useful to predict the maximum mechanism temperature, $T_{max}$, over a certain window of time, $t_{forecast}$. The optimal duration of $t_{forecast}$ will yield a definition of $T_{max}$ that is more easily predicted by $I_{mm}$, $I_{cm}$, and $E_{li}$. It is thus expected that the optimal duration of $t_{forecast}$ can maximize the correlation between the predictors $I_{mm}$, $I_{cm}$, and $E_{li}$ and the predictand $T_{max}$. In this study, an unsupervised data-driven anomaly detection algorithm utilizing an LSTM layer makes predictions of $T_{max}$ that minimize mean average error (MAE) in its training dataset, which is derived from thousands of observations during a single anomaly case.

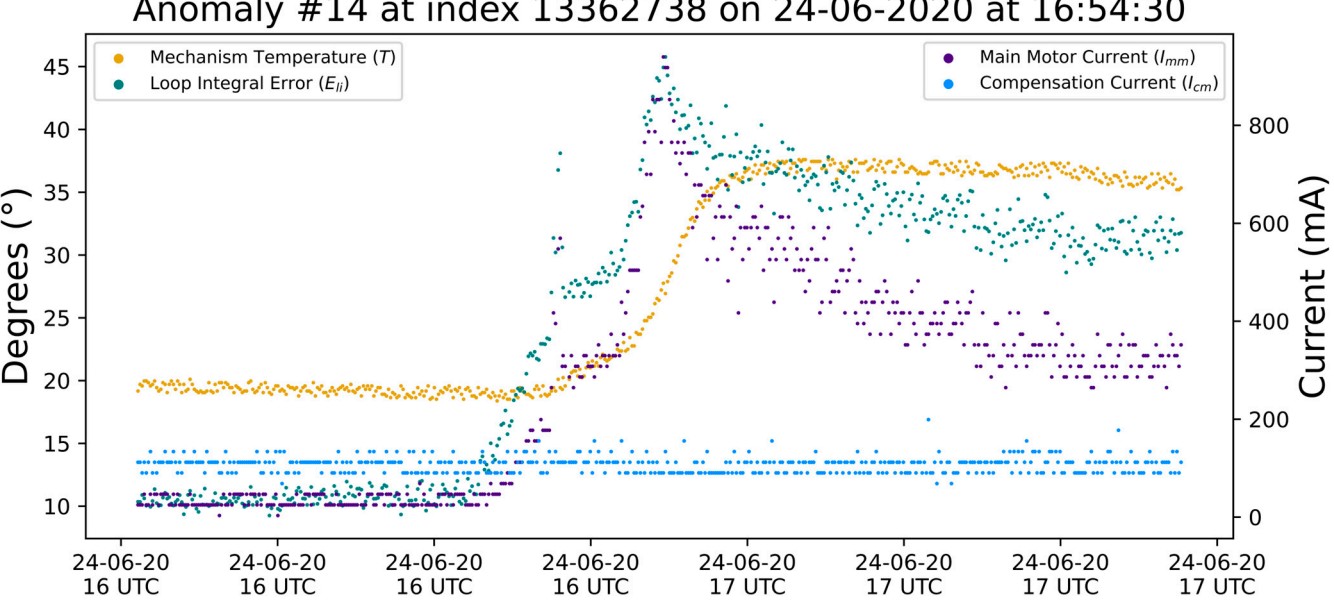

**Figure 5.** Time series of ATMS main motor current ($I_{mm}$), compensator motor current ($I_{cm}$), loop integral error ($E_{li}$), and mechanism temperature ($T$) during one of the cases included in this study.

To identify anomaly cases for this study, ICVS acquired S-NPP ATMS telemetry RDR data for the timespan of 8 December 2016 to 31 December 2021. From this time series, 17 instances where T exceeded 27 °C were identified. Of those 17 instances, only eight cases comprised at least 10 observations of T greater than 27 °C within 40,000 s of the first

observation over 27 °C, and of those eight cases, two of them were not unique, occurring within two weeks of a previous anomaly. To determine the optimal duration of $t_{forecast}$, ICVS computed the correlation between $T_{max}$ and each of the input parameters as a function of $t_{forecast}$ for each of these six cases over the full 80,000 s period. Detailed descriptions for the 6 cases are given in Section 3.2 below. In all six of these cases, the maximum correlation between $T_{max}$ and both $I_{mm}$ and $E_{li}$ occurred at either $t_{forecast}$ = 4 min or $t_{forecast}$ = 6 min. Consequently, the optimal predictand is $\hat{T}_{max}$ with a forecast window of 6 min. Training the LSTM meant picking one of the six anomaly cases to serve as a training dataset. Arbitrarily, the final anomaly, which occurred in November 2021, was chosen to train the model.

To train the neural network, all relevant telemetry parameters were normalized and scaled to the interval [0, 1]. In addition, the maximum of the input parameters over the previous 102 min was computed in order to reduce the influence of orbital fluctuations. In total, seven parameters were used as inputs to the model, recent single orbit maxima of $T$, $I_{mm}$, $I_{mm}$, and $E_{li}$, as well as the most recently observed $I_{cm}$, $I_{mm}$, and $E_{li}$.

For a perfect forecast of maxima in $T$ in the ensuing 6 min, $T_{max}$ is the "truth" against which the model's predictand, $\hat{T}_{max}$, can be compared. The temperature $T_{max}$ was first computed for 5400 observations preceding and succeeding the first moment when $T$ exceeded 27 °C for the training case in November 2021. Because ATMS telemetry RDR records data every 8 s, these 10,800 observations represented approximately one full day of observations. Unsupervised training began by assigning random weights to each neuron in the network and computing the predictand accordingly for each of the 10,800 points in the time series. To make computation easier, these 10,800 points in the time series were divided into smaller batches of just 64 observations each. Comparing the predicted values ($\hat{T}_{max}$) with the true ones ($T_{max}$), computing the mean absolute error (MAE), determining adjustments to neuron weights using gradient descent, and backpropagating those adjustments to the preceding layers constituted a single training iteration of the model, known as an epoch. To prevent the network from converging too quickly on a suboptimal solution, the learning rate was set to 0.0001. After 50 training epochs, the model consistently converged on a solution with a satisfactorily low MAE.

### 3.2. Case Studies

The rest of the five cases among the selected six anomalous ATMS RDR telemetry events were used to assess the performance of the derived coefficients from the training data set. For all five test cases, MAE across the 10,800 predictions in each test case consistently dropped to around 1 °C or below in multiple models trained independently on the same data. Such a low MAE bodes well for this forecasting technique. Results from all five cases are summarized in Table 1. Indeed, looking at individual case studies confirms the quality of the prediction. Furthermore, results comparing the predicted and measured (truth) maximum ATMS mechanism temperatures are presented in Figures 6–8 in several different ways. Specifically, Figure 6 shows the time series of the predictors ($I_{mm}$, $I_{cm}$, and $E_{li}$) and predictands ($T_{max}$ and $\hat{T}_{max}$) for each of the five cases, while Figure 7 shows a scatterplot depicting the relationship between the true predictand ($T_{max}$) and the forecasted predictand ($\hat{T}_{max}$). Figure 8 depicts the residual ($T_{max} - \hat{T}_{max}$) as a function of time for each case. Detailed descriptions per case are given below in reverse chronological order.

**Table 1.** Results from five testing case studies, one training case study, and whole time series.

| Date | N | Avg MAE (°C) | Avg RMSE (°C) |
|---|---|---|---|
| 15 January 2017 (Testing Data) | 89 | 1.00 | 1.09 |
| 20 February 2017 (Testing Data) | 89 | 1.11 | 1.26 |
| 21 September 2018 (Testing Data) | 89 | 0.68 | 1.33 |
| 25 February 2020 (Testing Data) | 89 | 1.07 | 1.37 |
| 24 June 2020 (Testing Data) | 89 | 0.86 | 1.13 |
| 18 November 2021 (Training Data) | 89 | 0.63 | 1.16 |
| Whole Series | 89 | 0.80 | 0.98 |

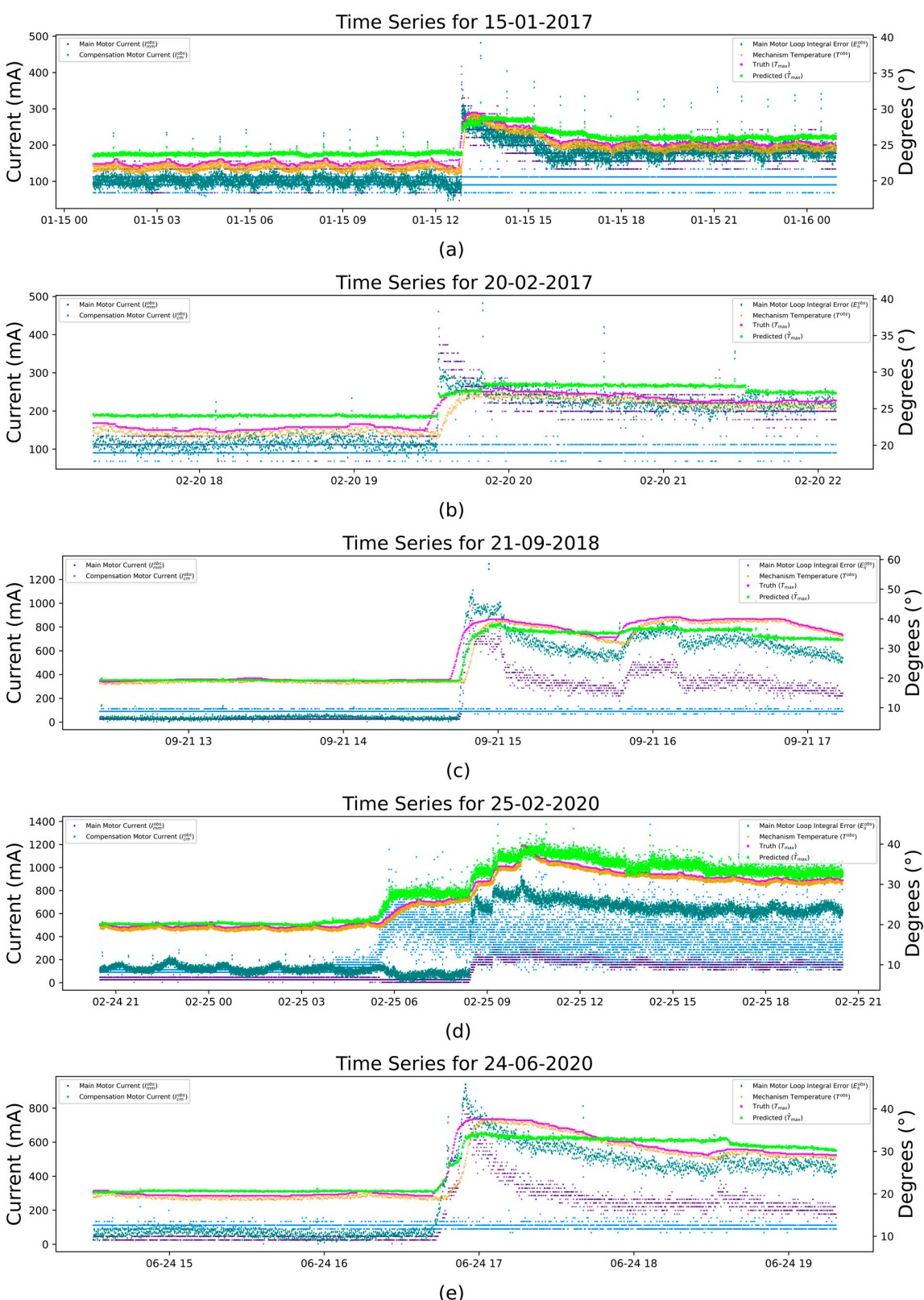

**Figure 6.** Time series showing the quality of the forecast for the temperature anomaly for five test cases that occurred on different dates: (**a**) 15-01-2017, (**b**) 20-02-2017, (**c**) 21-09-2018, and (**d**) 25-02-2020 (**e**) 24-06-2020.

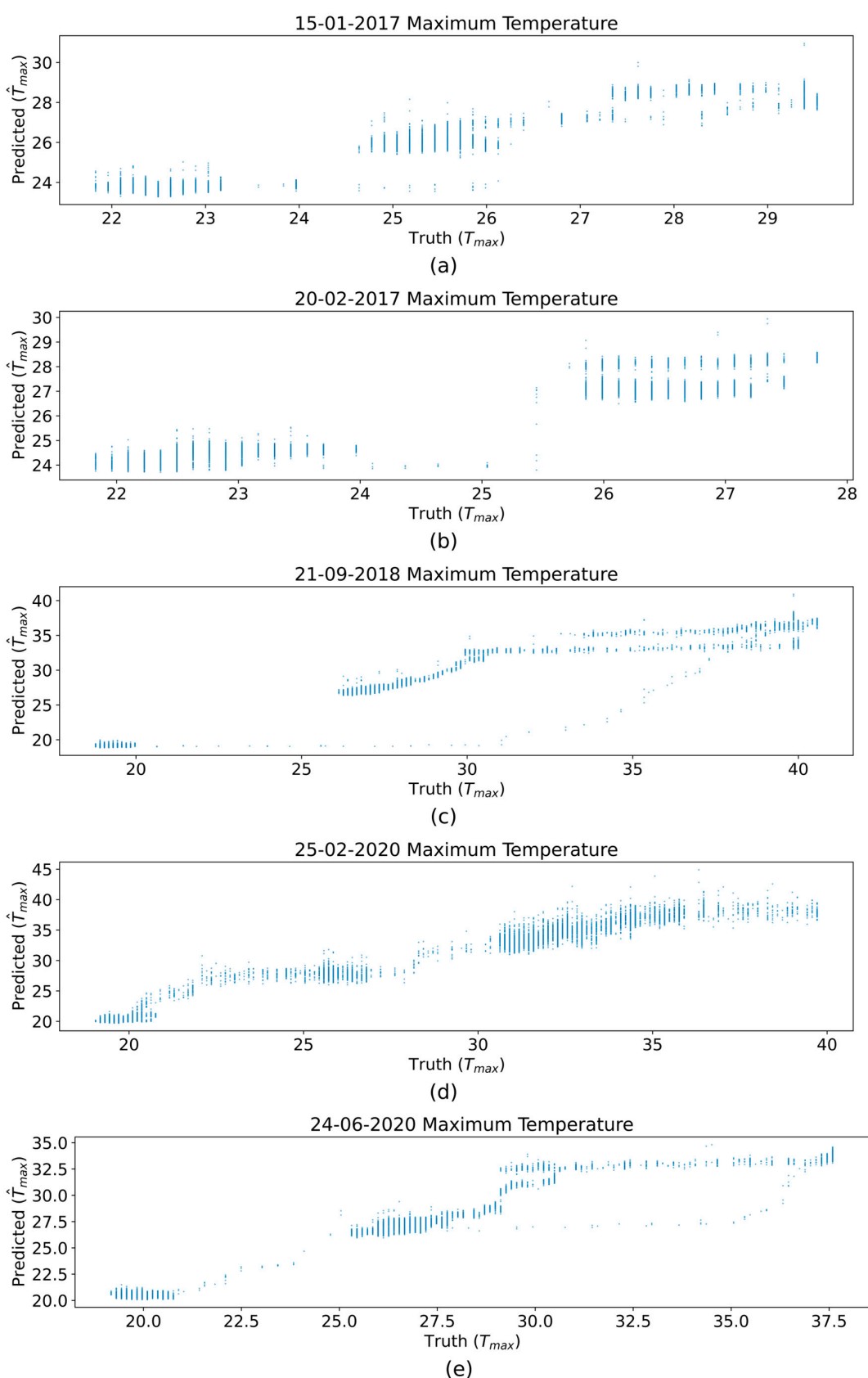

**Figure 7.** Same as Figure 6 except for scatterplots with the correlation between $T_{max}$ and $\hat{T}_{max}$ for five test cases: (**a**) 2017-01-15, (**b**) 2017-02-20, (**c**) 2018-09-21, (**d**) 2020-02-25, and (**e**) 2020-06-24.

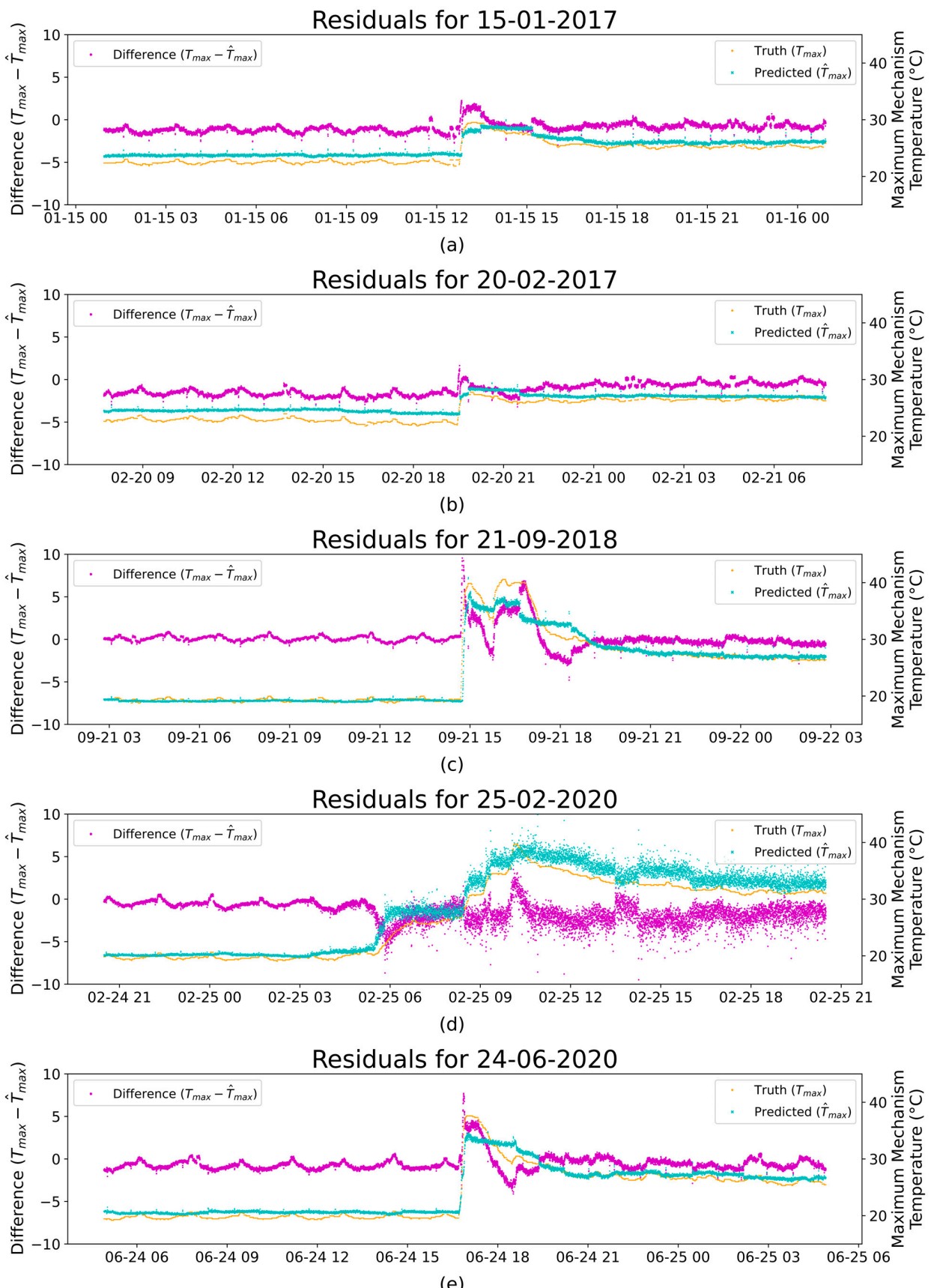

**Figure 8.** Same as Figure 6 except for the residuals of $\hat{T}_{max}$ for the five test cases: (**a**) 2017-01-15, (**b**) 2017-02-20, (**c**) 2018-09-21, (**d**) 2020-02-25, and (**e**) 2020-06-24.

For the anomaly case on 24 June 2020, it looks like the time series of the telemetry inputs ($I_{mm}$, $I_{cm}$, and $E_{li}$), the true predictand ($T_{max}$), and the forecasted predictand ($\hat{T}_{max}$) match quite well (Figure 6a). Both $T_{max}$ and $\hat{T}_{max}$ rise at nearly the same time and crest almost simultaneously a few minutes later. The MAE for this example is 0.987 °C, and the Pearson correlation coefficient between $T_{max}$ and $\hat{T}_{max}$ is 0.982 (Figure 7a). This was one of the larger anomalies examined in this study, and the model performed superbly.

The anomaly case on 25 February 2020 has a smaller magnitude than the 24 June 2020 anomaly, but the match between the true predictand ($T_{max}$) and the forecasted predictand ($\hat{T}_{max}$) appears to show a great deal of similarity. This anomaly features two significant increases in $T_{max}$, and from a subjective standpoint, the model predicts both adequately (Figure 6b). More rigorously, the Pearson correlation coefficient between $T_{max}$ and $\hat{T}_{max}$ is 0.988 (Figure 7b), which is the highest correlation observed across all five case studies. The MAE is 1.1 °C, which exceeds the MAE of the other cases but remains relatively small.

The case study on 21 September 2020 contains two $T_{max}$ spikes as well, but with a noticeable dip in between. Apparently, the forecasted predictand tracks the true predictand quite well (Figure 6c), which is confirmed by numerical measures. The Pearson correlation coefficient between $T_{max}$ and $\hat{T}_{max}$ is 0.981 (Figure 7c), and the MAE is 0.805 °C, which is the lowest MAE of all five case studies. The correlation scatterplot between $T_{max}$ and $\hat{T}_{max}$ shows a bifurcation, which is presumably a result of the dual peaks in the time series. If so, this may indicate that one spike was predicted more accurately than the other. Regardless, the trained neural network performed best when forecasting this case.

The results from the anomaly case on 20 February 2017 show similar patterns to all the other cases. Qualitatively, the forecasted predictand tracks the true predictand quite well (Figure 6d), and the statistical measures agree. The Pearson correlation coefficient between $T_{max}$ and $\hat{T}_{max}$ is 0.947 (Figure 7d), and the MAE was 1.084 °C. Accordingly, this modest anomaly was well predicted by the model.

A small anomaly on 15 January 2017 highlights similar patterns. At first sight, the forecasted predictand tracks the true predictand quite well (Figure 6e). Looking deeper at more quantitative measures, the Pearson correlation coefficient between $T_{max}$ and $\hat{T}_{max}$ is 0.964 (Figure 7e), and the MAE was 0.919 °C. Here, the model performed well in this case too.

Indeed, looking at individual case studies confirms the quality of the prediction. From the residuals of all five cases (Figure 8), there are brief periods when the model's forecast strays as much as 5 °C from the truth. Mostly, this stems from a bit of a delay between the increases in $T_{max}$ and $\hat{T}_{max}$. This may be a consequence of the fact that the optimal forecast period, $t_{forecast}$, is slightly different for $I_{mm}$ and $E_{li}$. Perhaps a better input into the forecast model can include time-lagged editions of either of these parameters rather than the contemporaneous observation of each along with their recent maximum. Nonetheless, these lags almost always err on the side of caution. Instances when $T_{max}$ exceeds $\hat{T}_{max}$ are vanishingly rare across all five case studies, which bodes well for the operational implementation of these forecasts onto the spacecraft flight software.

*3.3. Discussion for Small Anomalous Events*

The above analysis shows promising results with a high correlation coefficient between $T_{max}$ and $\hat{T}_{max}$ for all five cases. It is entirely feasible to predict ATMS motor maximum mechanism temperature using observations of $I_{mm}$, $I_{cm}$, and $E_{li}$ during a given time window and inputting them and their orbital maxima into an ANN featuring an LSTM layer. This is especially true for the cases with larger anomalies, as these cases exhibit some of the largest correlations between $I_{mm}$ and $T_{max}$, just like those found in the training data.

Smaller anomalies, on the other hand, tend to display much higher correlations with compensator current at the expense of smaller correlations with $I_{mm}$ and $E_{li}$ (Figure 9). A qualitative examination of smaller spikes in mechanism temperature (T) often shows similar increases in compensation motor current a few minutes earlier. Note that T does not increase by more than 5 °C and never exceeds 25 °C, which is 25 °C less than the

temperature where permanent damage to the instrument may occur. Because this pattern is not found in the training data, these smaller anomalies are not as easily predicted by the model. However, these anomalies are the least likely trigger an ATMS to enter safe mode because they are so tiny' thus, there is comparatively little importance to predicting them as accurately as the larger anomalies. Still, superior results might be achieved by training the model on major and minor anomalies separately, taking the maximum of the two forecasts into account when controlling the ATMS' safe mode status.

**Figure 9.** An example of a small increase in T that is correlated more strongly with $I_{cm}$ than $I_{mm}$.

All in all, the results from the LSTM neural network are very promising. Across all six unique instances when the gimbal mechanism sustained above 27 °C for more than 10 observations, the spike in temperature was accurately forecasted by a model trained on just one of those instances. By forecasting these temperature anomalies 6 min in advance, there should be sufficient warning to enter safe mode before permanent damage to the instrument can occur. Likewise, there is reasonable confidence that there is a low risk of damage whenever the forecast does not exceed 50 °C, even if the current temperature is above 40 °C. Consequently, these forecasts enable the instrument to continue collecting scientifically meaningful data even when the mechanism temperature is warm enough to trigger the current precautionary safe mode procedure.

## 4. Conclusions

This work demonstrates the ability of LSTM neural networks to forecast anomalies in time series in the ATMS telemetry data. The performance of the LSTM network presented in this study is quite good and reasonably anticipates all significant spikes in maximum mechanism temperature observed over a half-decade (2017–2022) aboard S-NPP ATMS. In all five test cases, the MAE was less than 11 °C, and the correlation between forecasts and truth was above 0.95. Harnessing LSTM predictions, it is thus feasible to build a robust near real-time (NRT) warning system that puts an ATMS into safe mode only when strictly necessary to prevent damage to the instrument. Importantly, using a neural network provides an outright improvement over existing methods for safeguarding the gimbal assembly against extremely high temperatures. Specifically, for all S-NPP temperature

anomalies from 2017 to 2021, the actual anomaly peak never exceeds the forecasted peak by more than 6 °C. Current procedures have the instrument enter safe mode if the temperature rises within 10 °C of the threshold at which permanent damage may occur, leading to fewer safe mode events and significantly decreasing the unnecessary loss of valid scientific SDRs. Because these accurate 6 min forecasts can be computed in a matter of seconds, implementation of this model into the flight control software aboard ATMS is entirely feasible. The use of this LSTM-based method is thus expected to preserve the collection of useful ATMS data even when the gimbal mechanism temperature rises to 40–44 °C.

Certainly, some limitations remain in the current LSTM method for predicting SNPP ATMS maximum mechanism temperature. The correlation between predictor and predictand reaches its maximum for lag times around 6 min. This short time window may be a concern for the ATMS instrument flight team seeking to execute ATMS safe mode in an operational flight software processing environment. In addition, only five large anomalous cases were tested in this study, and the algorithm was established using only SNPP ATMS data. Therefore, our future analysis will investigate the feasibility of a longer forecast period than 6 min without degrading the algorithm retrieval accuracy of the maximum mechanism temperature. The algorithm will be applied to more SNPP anomalous cases to ensure the stable performance of the LSTM method in predicting ATMS maximum mechanism temperatures. Comparisons of the performance of the LSTM method with that of other ANNs (e.g., an ordinary RNN) are another avenue for future study. Because spikes in the temperature increase in frequency as the scan drive mechanism ages, this time series is not stationary, and a Box-Jenkins model will not accurately predict future observations. Last but not the least important, the algorithm presented in this study will be applied to NOAA-20 and upcoming JPSS-2 ATMS anomalous mechanism temperature events in future studies. A reliable LSTM-based operational method is expected to be developed with those new analyses that can be applied to all ATMS telemetry RDR data to predict maximum mechanism temperatures with an optimal forecast time window.

**Author Contributions:** Conceptualization, N.S., W.D.P. and B.Y.; methodology, W.D.P.; software, W.D.P. and N.S.; validation, W.D.P.; formal analysis, W.D.P.; investigation, W.D.P. and N.S.; resources, B.Y.; data curation, N.S.; writing—original draft preparation, W.D.P.; writing—review and editing, B.Y.; visualization, W.D.P.; supervision, B.Y.; project administration, B.Y.; funding acquisition, B.Y. All authors have read and agreed to the published version of the manuscript.

**Funding:** This research is sponsored by the JPSS funding resource.

**Institutional Review Board Statement:** Not applicable.

**Informed Consent Statement:** Not applicable.

**Data Availability Statement:** All SDR/TDR datasets that support the findings of this study are openly available in NOAA CLASS at https://www.avl.class.noaa.gov/saa/products/catSearch, accessed on 5 May 2022.

**Acknowledgments:** The authors wish to thank the JPSS Program Office for supporting this work. manuscript contents are solely the opinions of the author(s) and do not constitute a statement of policy, decision, or position on behalf of NOAA or the U.S. Government. We thank two anonymous peer reviewers for their valuable comments and feedback.

**Conflicts of Interest:** The funders had no role in the design of the study; in the collection, analyses, or interpretation of data; in the writing of the manuscript; or in the decision to publish the results.

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
