# Peer review of "Forecasting Maximum Mechanism Temperature in Advanced Technology Microwave Sounder (ATMS) Data Using a Long Short-Term Memory (LSTM) Neural Network"

_atmosphere, doi:10.3390/atmos14030503_

Round 1

Reviewer 1 Report (Previous Reviewer 1)

The author solved my problem to some extent. Thank you for your revision. Only one question needs further correction. Please summarize the research gap and your work's contribution (or novelty) in the introduction section. In general, each contribution can be considered as the point-by-point response to the research gap. Therefore, it may be better to summarize these parts using a list (for example, a, b, c).

Author Response

Dear Reviewer,

            Thank for taking the time to review our manuscript and providing such helpful suggestions. With these comments taken into careful consideration, we have improved the current version of our manuscript by adding a concise list summarizing the existing research gaps. The authors welcome further comments, if necessary, but we are hopeful the current revision meets your high standards.

Sincerely,

Warren Porter

[email protected]

Scientific Analyst

Science Systems and Applications Inc.

Reviewer 2 Report (Previous Reviewer 3)

The authors considered all my suggestions, the manuscript can be published in this form. 

Author Response

Dear Reviewer,

            Thank for taking the time to review our manuscript and providing such helpful suggestions.

Sincerely,

Warren Porter

[email protected]

Scientific Analyst

Science Systems and Applications Inc.

This manuscript is a resubmission of an earlier submission. The following is a list of the peer review reports and author responses from that submission.

Round 1

Reviewer 1 Report

The paper titled “Forecasting Maximum Mechanism Temperature in Advanced Technology Microwave Sounder (ATMS) Data Using a Long Short-Term Memory (LSTM) Neural Network” seeks to improve upon existing methods and discover whether it is possible to predict spikes in mechanism temperature by using observations of various parameters found in ATMS telemetry RDRs in a Long Short-Term Memory (LSTM).

The research topic is interesting. However, the contribution of this study is still not clear. The research gaps and main contributions should be summarized and listed in the Introduction. It would be better for readers to catch your work’s novelty and contribution.

Benchmarks must be added.

Model parameters should be presented in detail.

In summary, the authors just apply the existing algorithms to one particular problem, but I don't see clearly how the algorithms are modified to tackle with it, or which novel techniques are developed. So I think that the paper requires a much better motivation, making it clear which is the novelty from a scientific point of view and how is such novelty achieved.

Reviewer 2 Report

This is a very good research work, which attempts to predict the maximum temperature of instruments using the LSTM neural network method. It is very meaningful to improve the detector performance. Although the results are good and the data supports the conclusions well, the pictures are not well drawn and the discussion section is inadequate. I recommend minor revision. The main modifications are as follows:

1. L41: 14 literatures are quoted in this sentence, which does not conform to the normal citation pattern. It is recommended that the author cite the necessary literatures.

2. L93 and L291: delete.

3. Figure 2, 3, 6, 7, and 8: The quality of these figures is so bad, please redraw them.

4. Table1 and Figure7: Besides MAE and correlation coefficient, it is suggested to add other evaluation index (e,g. RMSE)

5. This study lacks sufficient discussion. It is suggested that the author add a section 3.4 to compare the similarities and differences between the author’s study and other studies, so as to confirm the superiority of the author's research.

Reviewer 3 Report

Referee Report for “Forecasting Maximum Mechanism Temperature in Advanced Technology Microwave Sounder (ATMS) Data Using a Long Short-Term Memory (LSTM) Neural Network” submitted to Atmosphere.

The authors of the manuscript proposed LSTM methodology to forecast maximum mechanism temperature in ATMS. It is a simple application of a recurrent neural network but the methodology is applied in a perfect manner. It is an interesting paper and fits the readers of Atmosphere and can be published after some modification. Please consider the following.

In the abstract, there are some abbreviations but their definitions are missing. Please consider them.

In the introduction please mention the paper's novelty and compare the manuscript with the literature.

There are many possible neural network algorithms. The reason for choosing LSTM should be stated. As s future study idea, the authors may consider comparing the LSTM with other RNN-type algorithms.

In the data analysis part, did the authors compare the performance of the algorithm for different hyperparameter cases? If yes, please provide some evidence.

Since the data is a time series how do you comment on using the Box-Jenkins methodology? It may be discussed in the conclusion. 

Overall it is a well-designed manuscript. Good luck.